# The Key Role of Glutamate Dehydrogenase 2 (GDH2) in the Control of Kernel Production in Maize (*Zea mays* L.)

**DOI:** 10.3390/plants12142612

**Published:** 2023-07-11

**Authors:** Thérèse Tercé-Laforgue, Jérémy Lothier, Anis M. Limami, Jacques Rouster, Peter J. Lea, Bertrand Hirel

**Affiliations:** 1Institut Jean-Pierre Bourgin, Institut National de la Recherche Agronomique et de L’Environnement (INRAE), CEDEX, 78026 Versailles, France; ttlgdh@gmail.com; 2Univ Angers, Institut National de Recherche Pour L’Agriculture et L’Environnement (INRAE), Institut de Recherche en Horticulture et Semence (IRHS), 49007 Angers, France; jeremy.lothier@univ-angers.fr (J.L.); anis.limami@univ-angers.fr (A.M.L.); 3BIOGEMMA-LIMAGRAIN, Site de la Garenne, Route d’Ennezat, CS 90126, 63720 Chappes, France; jacques.rouster@limagrain.com; 4Lancaster Environment Centre, Lancaster University, Lancaster LA1 4YQ, UK; p.lea@lancaster.ac.uk

**Keywords:** glutamate dehydrogenase, heterozygous, homozygous, kernel yield, maize, metabolome, mutant

## Abstract

The agronomic potential of glutamate dehydrogenase 2 (GDH2) in maize kernel production was investigated by examining the impact of a mutation on the corresponding gene. *Mu*-insertion homozygous and heterozygous mutant lines lacking GDH2 activity were isolated and characterized at the biochemical, physiological and agronomic levels. In comparison to the wild type and to the homozygous *ghd2* mutants, the heterozygous *gdh2* mutant plants were characterized by a decrease in the root amino acid content, whereas in the leaves an increase of a number of phenolic compounds was observed. On average, a 30 to 40% increase in kernel yield was obtained only in the heterozygous *gdh2* mutant lines when plants were grown in the field over two years. The importance of GDH2 in the control of plant productivity is discussed in relation to the physiological impact of the mutation on amino acid content, with primary carbon metabolism mostly occurring in the roots and secondary metabolism occurring in the leaves.

## 1. Introduction

Significant progress has been made to better understand the physiological role of the mitochondrial glutamate dehydrogenase (NAD(H)-GDH, EC 1.4.1.2) both in model and crop plants. The enzyme operating at the interface of carbon (C) and nitrogen (N) metabolism has the potential capacity to assimilate inorganic N by combining ammonium with 2-Oxoglutarate to form glutamate. Glutamate dehydrogenase can also deaminate glutamate by means of a reversible reaction [1,2]:Glutamate + H_2_O + NAD^+^ <-------> 2-oxoglutarate + NH_3_ + NADH + H^+^

In a number of studies performed on several models and crop species, it has been proposed on a regular basis that GDH is able to assimilate ammonium under certain physiological conditions leading to a build-up of ammonium, e.g., following leaf protein hydrolysis during N remobilization or under various environmental stresses [3,4]. However, in other studies using mutants deficient for the enzyme activity, it has been shown that GDH operates in the reverse direction of glutamate deamination to release organic acids notably when the cell is C-limited [5,6]. Such finding agreed with previous labeling experiments which demonstrated that GDH deaminates glutamate [7] even when the activity of the enzyme is markedly increased in genetically modified plants [8].

In most plant species, two distinct genes encoding α and β GDH subunits have been identified so far, except in Arabidopsis [9] and rice [10], the two plant species in which a third gene or a fourth encoding was identified. In Arabidopsis, the third gene encoding a γ subunit is mostly expressed in roots [9], whereas in rice, the fourth gene is specifically expressed in floral organs [10]. These different GDH subunits are able to assemble apparently at random into active hexamers. Depending on the plant species, the level of expression of the different members of the *Gdh* multigene family and the relative proportion of the corresponding subunits, the isoenzyme pattern varies with plant organ, external stimuli such as N source, light regime [9,10,11], abiotic stresses [3] and plant internal signal molecules such as phytohormones [12]. However, the physiological significance of such environmental and species-specific variability in the GDH subunit composition is still not clearly understood as in most cases no clear relationship was observed between amino acid synthesis or degradation and GDH subunit composition irrespective of the plant developmental stage and the organ examined [13]. Nevertheless, there are strong lines of evidence that, at least in the leaves, the α and β subunits play a specific function at the interface of C and N metabolism by contributing to glutamate homeostasis and thus the subsequent synthesis and export of derived amino acids [14]. Glutamate is an important signaling molecule in higher plants [15], and as such, GDH could contribute to its homeostasis when there is a shortage of C, fitting with the mitochondrial location of the enzyme in the phloem tissues where an active translocation of organic molecules is occurring [16]. However, in Arabidopsis GDH-deficient mutants lacking the three isoenzymes, the plant phenotype was not visibly altered during plant development, suggesting that a compensatory mechanism circumvents the lack of enzyme activity [9].

As for several other enzymes involved in N assimilation and recycling, both transgenic studies and quantitative genetic approaches were then undertaken in order to determine if the reaction catalyzed by NADH-GDH is of major importance in the control of plant growth and productivity. The overexpression of an NADH-dependent enzyme from plants generally had a negative impact on growth and development [14,17]. Such negative impact was partly explained by modifications of the ratio of glutamate to glutamine, which was also dependent upon the increase in the activity of the α or β subunits [14]. To circumvent such negative impact, an NADPH-GDH gene originating from other organisms such as fungi or unicellular organisms was often used to increase the capacity of the plant to produce glutamate. Such genetic manipulation lead in several cases to an improvement in crop agronomic performances or stress resistance [18,19,20]. The improvement in plant performances was explained by the fact that heterologous NADP-GDH enzymes are able to catalyze ammonium assimilation at a low concentration, a metabolic function that does not occur when an NADH-dependent plant enzyme exhibiting a low affinity for ammonium is overexpressed in genetically modified plants [20].

Quantitative genetics studies in maize also suggested that GDH could be involved in the control of plant productivity, as colocalization of QTLs for yield and its components and grain metabolic efficiency were found with the enzyme activity [2,21]. However, these QTLs did not colocalize with the two structural genes *Gdh1* and *Gdh2* encoding the enzyme subunits located on chromosome 1 and 10, respectively (https://cur.maizegdb.org/zmdb.php, accessed on the 10 July 2023). As previously observed in other plant species, it is therefore likely that post-transcriptional modifications are involved in the control of the final enzyme activity [22,23]. More recently, a whole genome scan revealed that GDH could be an important locus associated with useful agronomic traits in durum wheat and thus could be used in new selection programs [24]. Interestingly, in an earlier study, it was proposed that modern maize hybrids did require a high GDH activity to maintain a high grain yield under non-limiting N fertilization conditions. Such hypothesis indicates that the level of the enzyme activity could be an important determinant in the control of maize agronomic performances [25]. Although maize mutants deficient for the GDH1 isoenzyme were isolated and showed increased sensitivity to low temperature, the impact of the mutation on both plant physiology and plant agronomic performances was not described [26].

Despite the information available, as outlined above, concerning the expression of the two GDH isoenzymes and the role of the enzyme in replenishing C molecules under certain physiological conditions in plants, the precise role of the two GDH isoenzymes in controlling plant growth and productivity needs to be further investigated. Using the maize Mutator (*Mu*) system, a mutant lacking GDH2 activity was isolated. The insertion lines of *Gdh2*::Mu have undergone extensive backcrossing in the wild type non-*Mu* line FV2 during which homozygous, heterozygous and null mutant lines were selected. In this paper, we have investigated the role of the GDH2 isoenzyme, by studying the properties of the insertion mutants at the molecular, biochemical and physiological levels and by examining the impact of the mutation on kernel yield and its components under agronomic conditions.

## 2. Results

### 2.1. Isolation and Characterization of a gdh2-Deficient Mutant

A *Mu*-insertion event within the gene encoding GDH2 was first identified using the collection of mutants established at Limagrain (formerly Biogemma SAS, Aubière, France, (https://www.limagrain-europe.com/en, accessed on the 10 July 2023). The *Mu*-insertion event within *Gdh2* was identified by the random sequencing of *Mu*-tagged fragments [27]. A maize line having an insertion of a transposable element between position chr10:135303729 and 135303730 (RefGen_V4,) [28] of the reference sequence in the gene encoding GDH2 (Zm00001d025984) was isolated. The allele thus obtained is named D0425. Sequence analysis indicated *Mu* had been inserted within exon 1 (Figure 1A,B). During the five rounds of backcrosses of the *gdh2*-deficient mutant in line FV2 and two rounds of self-pollination, no distorted segregation patterns were observed when monitoring with a PCR assay designed to detect heterozygous and homozygous plants for the insertion events in the gene encoding GDH2.

In order to determine if the insertion is in a homozygous or heterozygous form, three primers were defined according to the PCR-based KASP technology as shown in Figure 1A. An end-point fluorescence read and cluster analysis of the samples revealed VIC fluorescence for homozygous WT plants, FAM fluorescence for homozygous mutant plants and both VIC and FAM fluorescence for the heterozygous mutant plants (Figure 2).

### 2.2. Glutamate Dehydrogenase Protein Content and Isoenzyme Composition in the gdh2 Mutant

Protein gel blot analysis was first conducted to examine if the GDH protein content was modified in the leaves of the *gdh2* homozygous mutant. In the WT, two polypeptides of 41 and 42 kDa corresponding to the two GDH1 and GDH2 subunits [29] were detected with the antibodies raised against grape GDH [30]. In the mutant, the band of 42 kDa corresponding to the GDH2 protein was lacking (Figure 3A). In-gel activity staining was then used to detect the GDH isoenzyme composition in young leaves and roots of the *gdh2* mutant; only the most cathodal isoenzyme (GDH1), a homohexamer composed of six α-subunits, was present in comparison to the WT plant, in which the seven isoenzymes composed of heterohexamers between the polypeptides α and β in the WT were detected (Figure 3B). We also observed that, in both leaves and roots of the *gdh2* mutant line, the activity of the GDH1 isoenzyme was higher than in the WT. Such compensation mechanism has already been observed in Arabidopsis mutants lacking GDH2 activity [6] or in maize during hypoxia [31].

During the selection and backcrossing of the mutants, homozygous and heterozygous *gdh2* mutant lines were also obtained as shown in Figure 3C and Figure 4. One can observe that in the heterozygous mutant (m), the isoenzyme composition was slightly modified compared to the WT, leading to the disappearance of a number of cathodal isoenzymes due to the reduction in the amount of the β subunit which could not assemble with the most anodal isoenzyme represented by the α subunit of GDH1.

### 2.3. Metabolic Profile of the gdh2 Mutant

To evaluate the physiological impact of the *gdh2* mutation at the vegetative stage of plant development, homozygous and heterozygous mutant plants were grown under hydroponic conditions. Such growth conditions allowed for a comparative metabolomic analysis to be performed with the roots and leaves of the WT, because when plants were grown in the field, the analysis was not sufficiently reliable due to the difficulty of accessing the whole root system. The results of this study are presented in Table 1 and Table 2, respectively. Among more than 150 identified compounds, those showing statistically significant differences between the WT and the *gdh2* mutant (*p* ≤ 0.05) were mostly represented by amino acids in the roots, whereas in the leaves, C-containing molecules and secondary metabolites predominated. In the roots, the concentration of most amino acids was decreased by 15 to 25% in the *gdh2* heterozygous mutants, whereas in the homozygous mutant, such decrease was only observed for Alanine, Arginine, Glutamine and Threonine. Fewer C-containing metabolites such as Galactose and Ribose were detected in lower concentrations both in the homozygous and heterozygous mutants. *Myo*-Inositol was the compound exhibiting the highest decrease only in the heterozygous mutant. Digalactosylglycerol was the only metabolite present in a higher concentration in the two types of mutants.

In the leaves, the pattern of metabolite accumulation was much more complex, involving several classes of metabolites. Again, the most important changes were observed in the heterozygous *gdh2* mutant in which the concentration of C- and N-containing molecules was on average 25% lower. Among these molecules, Mannitol, Sorbitol, Arabitol 3-P, glycerate and Asparagine were those that are the most commonly detected in several plant species. An increase in a number of secondary metabolites mostly represented by phenypropanoids and their precursor was the main characteristic of the changes occurring only in the heterozygous mutant. However, such an increase was also observed in the homozygous mutant for a few of them, such as 3-*Trans*-Caffeoylquinate, 4-*Cis*-Hydroxycinnamate, *Trans*-Ferulate, Quinate and Shikimate. It was not possible to perform such a comparative study with plants grown in the field due both to the size of the root system and the difficulty in harvesting the plant material.

### 2.4. Mutant Phenotype and Biomass Production

To determine the impact of the homozygous and heterozygous mutation on plant phenotype and kernel production, plants were grown until maturity in the field over two years in 2010 and in 2013 under optimal N feeding conditions. Prior the transfer of the young plantlets to the field, the selection of the mutants was performed on the basis of their zymogram profile (see Figure 2C). As an example, Figure 4 illustrates the results obtained with the plants selected for the field trial performed in 2013.

At plant maturity, in comparison to the WT, an increase of 34% and of 38% in KY in 2010 and 213, respectively, was only observed in the heterozygous *gdh2* mutant (Table 3). Such an increase in KY was due to the increase in KN, whereas TKW remained unchanged. The phenotype of the ear in the WT and the homozygous and the heterozygous *gdh2* mutants is presented in Figure 5, showing that more kernels are present in the former. In the two field experiments, an increase in the heterozygous *ghdh2* mutant shoot dry matter production of 42% in 2010 and 26% in 2013 was also observed. In contrast, plant height in the WT and in the two types of mutants was similar (Table 3).

## 3. Discussion

Although a large number of studies have been devoted to deciphering the physiological role of the NADH-GDH in plants, its role with regards to notable plant growth and productivity in crops remains to be clearly assessed. Among these studies, the use of mutants and transgenic plants in which the enzyme activity was decreased or increased allowed for the demonstration of the fact that the two different subunits composing the enzyme seem to play a specific metabolic regulatory role, depending on their relative abundance when the enzyme is in the form of homohexamers or heterohexamers [14]. Such specific role was also highlighted when the metabolic profiling of Arabidopsis mutants lacking one or all GDH isoenzymes was studied [5,6]. However, these studies were performed on model plants, mostly because the availability and physiological characterization of mutants or transgenic plants altered for GDH activity in crops such as cereals or grain legumes remained rather limited [17,26]. Nevertheless, it was found in both maize and soybean that the balance between the two GDH isoenzymes could be involved in the control of plant biomass production [32].

Heterologous expression of an NADP(H)-dependent fungal enzyme in rice could improve both plant N assimilation and growth at least at the seedling stage [20]. As previously hypothesized following quantitative genetic studies [2,21], such a finding strengthened the idea that modulating GDH activity could be promising in order to improve crop productivity.

Generally, mutagenesis was a highly successful plant breeding strategy in order to improve crop productivity [33,34]. However, neither in a model nor in crop species has a beneficial impact in terms of NUE, yield of a mutation on a gene encoding a protein or an enzyme directly involved in N metabolism been reported so far. Only in maize did a dominant male-sterile mutant Ms44 encoding a lipid transfer protein show a reduced tassel growth and improved ear growth by partitioning more nitrogen to the ear, resulting in a 9.6% increase in kernel number when N is limited [35]. There are also other examples of beneficial impacts of a mutation on proteins involved in cytokinin response and on a NIN regulatory protein which led to the improvement of both NUE and plant productivity [36,37].

In the present investigation, we showed that in a maize heterozygous *gdh2* mutant, N metabolism was mostly modified in the roots. This modification was characterized by a decrease in the concentrations of almost all amino acids, while in the leaves, changes in N metabolism concerned only a few amino acids. The leaf Asparagine content was 55% lower, whereas that of Leucine, Cysteine and Tyrosine was 84, 61 and 21% higher. Moreover, it is likely that the upregulation of the shikimate pathway in leaves occurred at the expense of precursor amino acids such as Tryptophan, Phenylalanine and Tyrosine, leading to large accumulations of phenolic compounds such as 3-*trans*-Caffeoylquinate (3.23-fold increase) and of its precursor quinate that showed a 1.96-fold increase. Altogether, these changes were accompanied by an increase in grain yield when compared to the WT or to the homozygous mutant plants.

The major importance of the root system, with respect to the role of the high NADH-GDH activity, was previously revealed by means of metabolome and transcriptome analyses of Arabidopsis mutants lacking any enzyme activity [6]. However, the finding that these changes are mostly occurring in the heterozygous maize *gdh2* and not when GDH2 activity is lacking in the homozygous mutant is puzzling. It has been previously shown that overexpressing the two genes encoding GDH (GdhA and GdhB) individually or simultaneously in tobacco plants induced a differential accumulation of amino acids, such as glutamate and glutamine, notably when one of them (*GdhA*) was overexpressed [14]. Similar results were obtained when, instead of a plant enzyme, a bacterial enzyme was overexpressed in tobacco [38]. Although these studies were conducted in a dicot species, it seems logical to observe that when there is a reduction in the activity of one of the two maize GDH isoenzymes, glutamine and glutamate and a number of derived amino acids are present in lower concentrations. It is therefore attractive to think that metabolic changes are only occurring when GDH activity decreases but only when this decrease reaches a certain threshold. Such metabolic changes have been previously observed in Arabidopsis mutants defective in two of the three genes encoding the enzyme. They were mainly characterized by strong perturbation in the accumulation of most amino acids, notably during prolonged dark conditions [5]. Although these metabolic perturbations appear to be variable according to the species-specific GDH isoenzyme complement, it confirms that the enzyme plays a major role at the interface of C and N metabolism by controlling the level of glutamate and glutamine, both molecules being the precursors of most of the other amino acids [8].

It was more puzzling to observe that both grain and biomass production were enhanced from 26 to 42% only in the *gdh2* heterozygous mutant. At this stage of our investigation, one can hypothesize that more amino acids are exported from the roots in order to provide the N necessary to fill the grain [39]. In the leaves, the large increase in Cysteine, consistent with the decrease in its precursor 3-P-Glycerate which could be related to a better assimilation of sulfur and the increase in Leucine, an important amino acid for the structure of Leucine-rich repeat receptor-like kinases [40] and transcription factors such as Leu zippers [41], could also contribute to a better development of the heterozygous *gdh2* mutants. It is more difficult to explain the accumulation of several secondary metabolites only in the leaves of the heterozygous mutant. In several studies, it has been observed that secondary metabolite accumulation in maize leaves is often correlated with grain yield both in lines and hybrids [42,43]. The accumulation of these classes of metabolites in the leaves is also modified when ammonium assimilation is altered in glutamine synthetase-deficient mutants [44] and when plant N availability is modified during plant vegetative development [45,46]. Further work will be necessary to identify the link between the activity of the different GDH isoenzymes and secondary metabolism and their impact on plant productivity. Interestingly, Mungur et al. [38] observed that the relative proportions of several alkaloids and phenolics are strongly modified in transgenic tobacco plants overexpressing GDH from *E. coli*. Digalactosylglycerol, a chloroplastic glycolipid, which is the only metabolite exhibiting an increase in the roots of the heterozygous mutant, could also be an important molecule involved in the control of yield and its components [47]. Interestingly, transgenic plants constitutively expressing *Gdh1* and *Gdh2* sequences in the antisense orientation were unable to produce seeds. This finding suggests that, unlike in Arabidospis, the lack of enzyme activity in maize has strong repercussions on its agronomic performances. In line with this observation, we were not able to isolate a maize knock out GDH1 mutant, suggesting that the GDH1 isoenzyme is probably essential for plant development; although, in a previous report, it was observed that the mutants are phenotypically indistinguishable from the WT [26].

## 4. Materials and Methods

### 4.1. Isolation and Characterization of the gdh2 Mutant

A mutator (*Mu*)-insertion event within the gene encoding *Gdh2* was identified using the collection of mutants in the FV2 line background produced at Biogemma (Chappes, France). The *Mu*-insertion event within *Gdh2* was identified by the random sequencing of Mu-tagged fragments as described by Hanley et al. [27]. Sequence analysis indicated that *Mu* had been inserted within exon 1 (Figure 1A). During the five rounds of backcrosses, crosses and two selfings, no distorted segregation patterns were observed when monitoring with a PCR assay designed to detect heterozygous and homozygous plants for Gdh2 insertion events. In order to determine if the insertion is in a homozygous or heterozygous form, three primers were defined according to the PCR-based KASP technology: one allele-specific forward primer of the GDH2 sequence (named D0425_EPF_F04_vic: ATCGAAGCTGCTCGGCCTC) with a proprietary tail sequence corresponding with VIC dye, one allele-specific forward primer of the endogenous transposable element (named O*Mu*A_G_fam: CTTCGTCCATAATGGCAATTATCTCG) with a proprietary tail sequence corresponding with FAM dye and a third common allele-specific reverse primer of the gene encoding GDH2 (named D0425_EPF_R04: AGACGCCACAAGCAACACG). These three primers were used simultaneously in a PCR amplification experiment using the Kaspar protocol from LGC Genomics, Teddington, Middlesex, UK, starting with genomic DNA extracted using the QIAGEN DNeasy Plant Kit (Qiagen Sciences, Germantown, MD, USA) in which sodium metabilsulfite was included [48]. Amplification was conducted using 50 cycles (94 °C 15 min, 94 °C 10 s, 57 °C 20 s and 72 °C 40 s in a mix containing 1.8 mM MgCl_2_). An end-point fluorescence read and cluster analysis of the samples revealed VIC fluorescence for homozygous WT plants, FAM fluorescence for homozygous mutant plants and both VIC and FAM fluorescence for the heterozygous mutant plants. The WT plants were represented by 24 plants, the homozygous mutants by 6 plants and the heterozygous mutants by 14 plants, each replicated twice.

### 4.2. Plant Material for Molecular, Physiological and Agronomic Studies

Seeds of the homozygous and heterozygous *gdh2* mutant lines and hybrids and the corresponding WT were first sown on coarse sand, and after 1 week, when the 6th leaf had emerged, they were transferred to hydroponic culture for root and shoot harvesting. For the hydroponic culture, 21 plants (7 for the WT and 7 for the heterozygous and homozygous mutants) were randomly placed on a 130 L aerated culture unit. The experiment was performed in triplicate for each line and plants were grown for 15 days in a growth chamber with a 16/8 light/dark period. A photosynthetic photon flux density of 400 μmol·m^−2^·s^−1^ was provided by LED lamps (Led Power, Saint-Calais, France). The relative humidity was maintained at 60% saturation. Plants were harvested at the 7–8 leaf stage between 9 and 12 a.m. and separated into shoots and roots. The shoot and root samples were immediately placed in liquid N_2_ and then stored at −80 °C until further analysis.

For the field experiments, the *gdh2* homozygous and heterozygous mutants and WT lines were grown in the field at INRA, Versailles, France (N 48°48.133′, E 2°04.942′) in deep silt loam without any stone. The number of plants was variable for the *gdh2* homozygous and heterozygous mutants and WT lines depending on seed availability. In 2011: 3 homozygous mutant plants, 7 heterozygous mutant plants and 11 WT; in 2013: 8 homozygous mutant plants, 4 heterozygous mutant plants and 12 WT. Three replicates were grown for each for each mutant and WT plant. The level of N fertilization was 175 kg/ha and N provided by the soil was estimated at 60 kg/ha. Both phosphorus (P205) and potassium (K20) were also applied at 100 kg/ha. The two types of mutant lines and the WT were grown in one row with a border row (line MBS857) between each and outside of the three rows. Line MBS was also used to complete the mutant rows to a number of plants comparable to that of the WT. The plants were sown on 10 May 2010 and on 20 May 2013. In the 2010 experiment, from the batch of seeds (named EB-07S-A-00053), 3 homozygous, 7 heterozygous and 11 WT plants were first selected following an in-gel assay for GDH activity as described in Figure 2C and then sown. In the 2013 experiment, it was 8 homozygous, 4 heterozygous and 12 WT plants. Agronomic traits used to evaluate the plants were plant height, shoot DW, Grain Yield (GY) and its components: Kernel Number/plant (KN) and Thousand Kernel Weight (TKW). For more details about the procedure used to measure the agronomic traits, see Bertin and Gallais [49,50].

### 4.3. Enzymatic In Vitro and In-Gel Assay, Determination of Total Soluble Protein and Protein Gel Blot Analysis

Soluble proteins were extracted from frozen leaf and root material harvested from plants grown under hydroponic conditions and stored at −80 °C. All extractions were performed at 4 °C. Glutamate dehydrogenase (NAD(H)-GDH) was measured as described by Turano et al. [51]. In-gel detection of GDH-NAD-dependent activity was performed as described by Restivo [22]. As previously shown by Loulakakis and Roubelakis-Angelakis [52], staining of NADH-GDH activity revealed the same isoenzyme profile (data not shown). However, in the present study, NAD-GDH in-gel detection was used because of its higher sensitivity. Soluble protein was determined using a commercially available kit (Coomassie Protein assay reagent, Biorad, München, Germany) using bovine serum albumin as a standard. Leaf and stem soluble proteins were transferred to nitrocellulose membranes for Western blot analysis and polypeptide detection was performed using polyclonal antiserum raised against GDH of grape leaf [26].

### 4.4. Metabolome Analysis

For the leaf and root metabolome analyses of the WT and the *gdh2* mutant, all steps were adapted from the original protocol described by Fiehn [53], following the procedure described in Amiour et al. [46]. The ground frozen leaf samples (25 mg fresh weight) were resuspended in 1 mL of frozen (−20 °C) water: chloroform:methanol (1:1:2.5) and extracted for 10 min at 4 °C with shaking at 1400 rpm in an Eppendorf Thermomixer. Insoluble material was removed by centrifugation and 900 µL of the supernatant was mixed with 20 µL of 200 µg/mL ribitol in methanol. Water (360 µL) was added and, after mixing and centrifugation, 50 µL of the upper polar phase was collected and dried for 3 h in a Speed-Vac and stored at −80 °C. For derivatization, samples were removed from −80 °C storage, warmed for 15 min before opening and Speed-Vac-dried for 1 h before the addition of 10 µL of 20 mg/mL methoxyamine in pyridine. The reactions with the individual samples, blanks and amino acid standards were performed for 90 min at 28 °C with continuous shaking. An amount of 90 µL of N-methyl-N-trimethylsilyl-trifluoroacetamide (MSTFA) was then added and the reaction continued for 30 min at 37 °C. After cooling, 50 µL of the reaction mixture was transferred to an Agilent vial for injection. For the analyses, 3 h and 20 min after derivatization, 1 µL of the derivatized samples was injected in the Splitless mode onto an Agilent 7890A gas chromatograph (GC) coupled to an Agilent 5975C mass spectrometer (MS). The column used was an Rxi ^®^-5Sil MS from Restek (30 m with 10 m Integra-Guard column). The oven temperature ramp was 70 °C for 7 min, then 10 °C/min up to 325 °C, which was maintained for 4 min. For data processing, Raw Agilent datafiles were converted into the NetCDF format and analyzed with AMDIS (http://chemdata.nist.gov/dokuwiki/doku.php?id=chemdata:amdis, accessed on the 10 July 2023). Peak areas were then determined using the quanlynx software v 4.0(Waters) after conversion of the NetCDF file into the masslynx format. Statistical analyses were carried out with TMEV http://www.tm4.org/mev.html, accessed on the 10 July 2023. Univariate analyses by permutation (1-way ANOVA and 2-way ANOVA) were first used to select the metabolites exhibiting significant changes in their concentration (*p* ≤ 0.05). Amino acid standards were injected at the beginning and end of the analyses, for the monitoring of derivatization stability. An alkane mixture (C10, C12, C15, C19, C22, C28, C32, C36) was injected in the middle of the run for external retention index (RI) calibration. For the analysis of the leaf samples, metabolite standards were injected at the beginning and end of each analysis. The metabolite concentration is expressed as nmol mg^−1^ leaf FW.

### 4.5. Statistics

For the metabolome analysis, results are presented as mean values for six plants. The significance of the differences between mean values was determined by Student’s *t*-test.

## 5. Conclusions

Although our knowledge on the physiological role of the enzyme GDH has been greatly improved over the last two decades, notably concerning its function at the interface of C and N metabolism, further work is still required to demonstrate if the enzyme plays a key role in crop productivity. In the present investigation, we showed that important changes in the pool of amino acids in the roots and in the secondary metabolites content in the leaves are only occurring in *gdh2* heterozygous mutants lacking one of the two isoenzymes composing the heterodimeric protein. More interestingly, we also observed that *ghd2* heterozygous mutants produced more kernels when grown in the field, opening interesting perspectives towards future agronomic applications.

## Figures and Tables

**Figure 1 plants-12-02612-f001:**
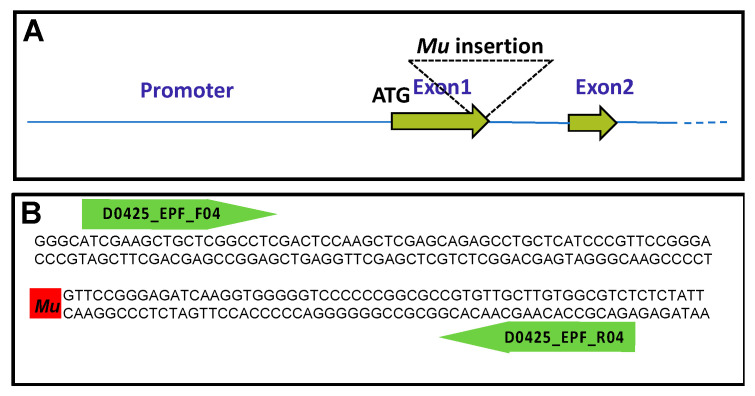
Characterization of the *gdh2*::*Mu*-insertion event. (**A**) Insertion position of the mutator element (*Mu*) within the *gdh2* gene. The *gdh2* gene structure and exon sizes were determined by sequencing genomic DNA PCR products using primers designed from the corresponding cDNA sequences [29]. The *gdh2* gene consists of 9 exons 9 introns. Only the positions of the ATG start codon (1 bp), Exon1, Exon 2 and the promoter region are shown. (**B**) The red box indicates the positions of the *Mu* element located at the end of exon 1 within which the primer O*Mu*A_G_ was used together with the two primers D0425_EPF_FO4 and D0425_EPF_RO4 to identify homozygous and heterozygous mutants.

**Figure 2 plants-12-02612-f002:**
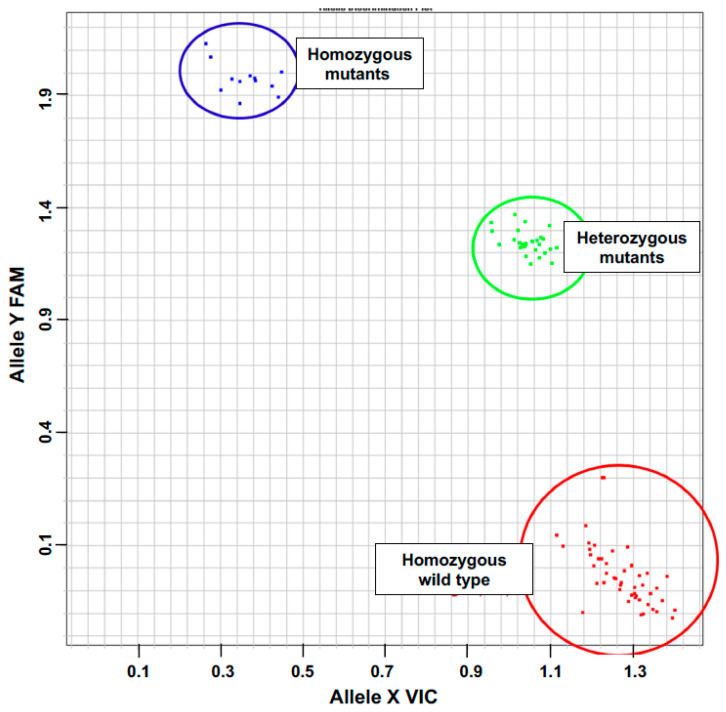
Identification of homozygous and heterozygous mutants using the Kaspar technology. DNA was extracted from individual plants obtained following five backcrosses and two selfings and amplified using the *Mu*-element primer O*Mu*A_G_ together with the two *Gdh2* gene-specific primers D0425_EPF_FO4 and D0425_EPF_RO4 shown in Figure 1. An end-point fluorescence read and cluster analysis of the samples revealed VIC fluorescence for homozygous WT plants (red), FAM fluorescence for homozygous mutant plants (blue) and both VIC and FAM fluorescence for the heterozygous mutant plants (green).

**Figure 3 plants-12-02612-f003:**
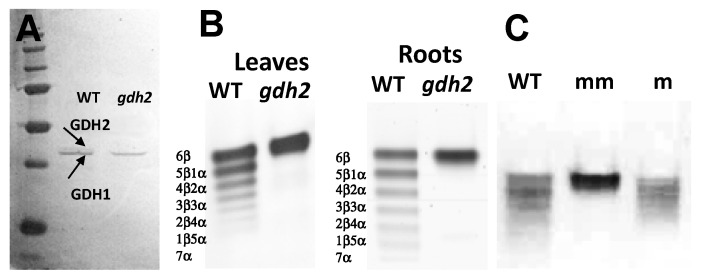
GDH isoenzyme composition in the *gdh2* mutant line. (**A**) Protein gel blot analysis of the GDH subunit composition in leaves of the WT and the *gdh2* mutant line using antibodies raised against the enzyme from grapevine [30]. The upper band (molecular mass of 42 kD) corresponds to the GDH2 subunit, and the lower band (molecular mass of 41 kD) corresponds to the GDH1 subunit. On the left side of the panel, the position of the protein molecular mass markers is shown (170, 113, 94, 52, 35, 29 and 21 kD from top to bottom). (**B**) Protein extracts of leaves and roots of the wild type (WT) and the *gdh2* mutant lines were subjected to native PAGE followed by NAD-GDH in-gel activity staining. The position of GDH1 and GDH2 homohexamers and of the different subunit combinations of the seven isoenzymes α and β detected in the WT are indicated in panels (**B**,**C**), respectively. (**C**) NAD-GDH isoenzyme patterns in the *gdh2*-deficient mutant. Leaf soluble protein extracts of heterozygous (m) and homozygous (mm) mutant lines and the WT.

**Figure 4 plants-12-02612-f004:**
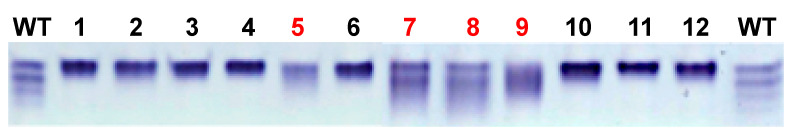
NAD-GDH isoenzyme patterns in the *gdh2*-deficient mutant grown in the field in 2013. Leaf soluble protein extracts of heterozygous (5, 7, 8 and 9) and homozygous (1, 2, 3, 4, 10, 11 and 12) mutant lines and the WT. Zymograms were performed on young leaf samples harvested at the 6-leaf stage.

**Figure 5 plants-12-02612-f005:**
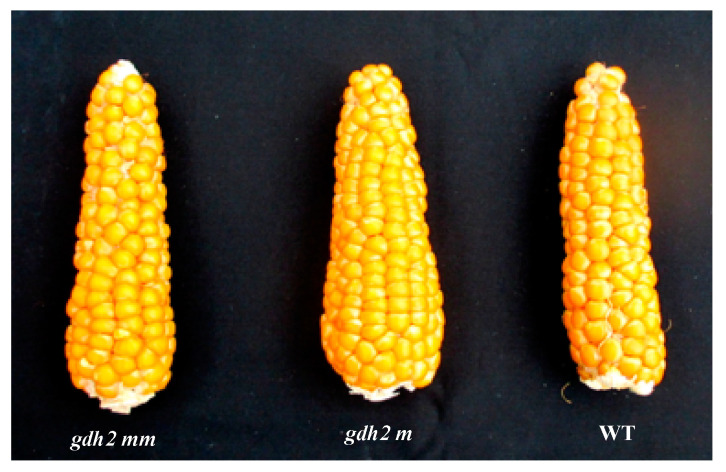
Phenotype of the ear in GDH2-deficient mutant lines. Ears of WT, homozygous (mm) and heterozygous (m) *gdh2* mutants of maize in line LMPD72 harvested at maturity and grown under non-limiting N conditions in the field at Versailles in 2010.

**Table 1 plants-12-02612-t001:** Average root metabolite content of the maize heterozygous and homozygous *gdh2* mutants. Mean data are expressed in nmol mg^−1^ leaf FW^−1^. Seven individual plants grown under hydroponic conditions were analyzed. The *t*-test indicates significant differences in the amount of metabolite between the wild type (WT) and the heterozygous (m) and homozygous (mm) mutant. FC: fold change between the WT and the two types of mutants. nc: no change between the WT and the mutant.

Heterozygous					Homozygous		
Metabolite	WT	m	FC	*t*-Test	mm	FC	*t*-Test
Alanine	0.3140	0.2356	0.75	0.0002	0.2430	0.77	0.0403
Allantoin	0.0019	0.0006	0.57	0.0313	nc	nc	nc
Arginine	0.0437	0.0217	0.51	0.0000	0.0246	0.58	0.0014
Asparagine	0.1336	0.0692	0.59	0.0062	nc	nc	nc
Aspartate	0.2034	0.1643	0.81	0.0018	nc	nc	nc
Glutamate	0.6834	0.5697	0.84	0.0343	nc	nc	nc
Glutamine	0.4184	0.3329	0.82	0.0495	0.3310	0.81	0.0426
Homoserine	0.0002	0.0001	0.85	0.0593	nc	nc	nc
Leucine	0.0314	0.0275	0.87	0.0046	nc	nc	nc
Lysine	0.0091	0.0058	0.69	0.0230	nc	nc	nc
Phenylalanine	0.0174	0.0146	0.84	0.0278	nc	nc	nc
Threonine	0.0467	0.0393	0.85	0.0205	0.0393	0.85	0.0512
Tryptophan	0.0052	0.0042	0.84	0.0698	nc	nc	nc
Tyrosine	0.0500	0.0427	0.86	0.0133	nc	nc	nc
Galactose	0.0081	0.0064	0.81	0.0220	0.0065	0.82	0.0399
*Myo*-Inositol	0.0209	0.0136	0.66	0.0001	nc	nc	nc
Ribose	0.0274	0.0226	0.84	0.0421	0.0219	0.82	0.0251
Digalactosylglycerol	0.0010	0.0013	1.27	0.0078	0.0015	1.51	0.0129

**Table 2 plants-12-02612-t002:** Average leaf metabolite content of the maize heterozygous and homozygous *gdh2* mutants. Mean data are expressed in nmol mg^−1^ leaf FW^−1^. Seven individual plants grown under hydroponic conditions were analyzed. The *t*-test indicates significant differences in the amount of metabolite between the wild type (WT) and the heterozygous (m) and homozygous (mm) mutant. FC: fold change between the WT and the two types of mutants. nc: no change between the WT and the mutant.

Heterozygous					Homozygous		
Metabolite	WT	m	FC	*t*-Test	mm	FC	*t*-Test
Arabitol	0.0052	0.0039	0.76	0.0014	nc	nc	nc
Asparagine	0.1695	0.0308	0.45	0.0501	nc	nc	nc
α-Amyrin	0.0020	0.0011	0.64	0.0293	nc	nc	nc
Erythronate	0.0004	0.0002	0.78	0.0424	nc	nc	nc
Ethanolamine	0.0315	0.0235	0.75	0.0004	0.0260	0.83	0.0260
Galactosylglycerol	0.0008	0.0005	0.63	0.0027	0.0006	0.76	0.0399
3-P-Glycerate	0.0013	0.0008	0.65	0.0055	nc	nc	nc
Mannitol	0.0030	0.0021	0.69	0.0002	nc	nc	nc
*Myo*-Inositol	0.0041	0.0032	0.83	0.0529	0.4543	1.22	0.0341
Sorbitol	0.0039	0.0030	0.80	0.0629	nc	nc	nc
1-3-Diaminopropane	0.0002	0.0003	1.34	0.0490	nc	nc	nc
3-*trans*-Caffeoylquinate-	0.0003	0.0006	3.23	0.0077	0.0001	1.28	0.0738
4-*cis*-Hydroxycinnamate	0.0001	0.0001	1.18	0.0595	0.0004	1.42	0.0234
Caffeate	0.0023	0.0028	1.26	0.0744	nc	nc	nc
Citrate	0.4600	0.5606	1.24	0.0165	nc	nc	nc
Cystein	0.0338	0.0450	1.61	0.0371	nc	nc	nc
Dopamine	0.0001	0.0002	1.38	0.0477	nc	nc	nc
*Trans*-Ferulate	0.0001	0.0002	1.74	0.0345	0.0003	2.39	0.0342
Leucine	0.0061	0.0101	1.84	0.0407	nc	nc	nc
Phytol-2	0.0005	0.0006	1.36	0.0148	nc	nc	nc
Quinate	0.0322	0.0586	1.96	0.0012	0.0452	1.51	0.0146
Shikimate	0.0285	0.0510	1.89	0.0010	0.0391	1.45	0.0224
Tyrosine	0.0189	0.0223	1.21	0.0707	nc	nc	nc

**Table 3 plants-12-02612-t003:** Agronomic performances of maize mutants deficient in the gene encoding GDH2 isoenzyme.

	Plant Height (cm)	Shoot DW (g)	KY (g)	KN	TKW (g)
Year 2010					
WT	181 ± 1.6	51.4 ± 2.6	29.9 ± 1.6	189 ± 10	162 ± 14.3
*gdh2* (m)	181 ± 3.5	73 ± 5.9 (42) ^a^	40 ±5.4 (34) ^a^	247 ± 15 (30) ^a^	158 ± 13
*gdh2* (mm)	180± 4.8	63 ± 5.8 (23)	26.6 ± 0.8	195 ± 3	138 ± 5
Year 2013					
WT	137± 3.6	84 ± 7	36.6 ± 5.1	156 ± 21	237 ± 7.7
*gdh2* (m)	141 ± 5.5	106 ± 7.4 (26) ^a^	50.4 ± 2.6 (38) ^a^	216± 12 (38) ^a^	235 ± 8.0
*gdh2* (mm)	131 ± 5.4	73.0 ± 10	31.5 ± 4.2	133 ± 20	246 ± 33

Each value is the mean ± SE obtained from homozygous (mm) and heterozygous (m) mutant lines and WT lines in 2010 and 2013. Plants were grown in the field as described in Materials and Methods. The increased values indicated in parentheses are expressed as a percentage of the value in the WT. ^a^ Significantly different from the WT at 0.05 probability level.

## Data Availability

Not available.

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
