# Peer review of "The Key Role of Glutamate Dehydrogenase 2 (GDH2) in the Control of Kernel Production in Maize (Zea mays L.)"

_plants, 2023, doi:10.3390/plants12142612_

Round 1
Reviewer 1 Report
Is a really nice paper, written by very well known experts in the field on nitrogen metabolism and photorespiration. In the manuscript, the agronomic potential of glutamate dehydrogenase 2 (GDH2) in maize kernel was investigated by examining the phenotype of a targeted mutant. Very interestingly, a substantial increase in kernel yield was found in the heterozygous mutant after two different field trials in different years.
The article is well written, the experiments well conducted and the conclusions are supported by the experimental data. This manuscript add a more layer of complexity to the "GDH puzzle". I definitely reccomend it for pubblication, I found no flaws or points that need to be improved.
Author Response
We thank reviewer 1 for the very positive comments on our work.
Reviewer 2 Report
The MS "The key role of glutamate dehydrogenase 2 (GDH2) in the con- 2
trol of kernel production in maize (Zea mays L.)" did a good work, I suggest to publish it in Plants.
1. In fig3B, there is GDH2 in the gdh2 in Roots, but not in the leaves, please check. Enlarge the image to make it clearer 。
2.In table 1 and 2, what about the SD.
3.In fig4, the WT is not consistent, please check, and lengthen the image to make it clearer.
Author Response
The MS "The key role of glutamate dehydrogenase 2 (GDH2) in the control of kernel production in maize (Zea mays L.)" did a good work, I suggest to publish it in Plants.
We thank reviewer 1 for the very positive comment on our work.
- In fig3B, there is GDH2 in the gdh2 in Roots, but not in the leaves, please check. Enlarge the image to make it clearer.
- The entire figure has been modified and its quality improved as also requested by reviewer 3. Only the the subunit composition of the different GDH isoenzymes have been indicated on the figure to avoid any confusion.
2.In table 1 and 2, what about the SD.
- As a t-test statistical analysis has been conducted to identify significant differences between the WT and the mutant (indicated in the Table), we did not add the values for SD in order to simplify the table as much as possible to make it easy to read like in most our previous publications presenting such kind of data.
3.In fig4, the WT is not consistent, please check, and lengthen the image to make it clearer.
- We agree that Figure 4 is improvable. To make it clearer, we have removed the three WT samples in which the separation of the different subunits was not good enough due to the wideness of the original gel. All the other samples are identical to those shown in the previous version. We have also improved the quality of the figure.
Reviewer 3 Report
1. I think it would be useful to include more information in the introduction about GDH genes in general in different plant species: for example is A. thaliana the only species with 3 genes, what are the differences in expression patterns in maize and other species, at the amino acid level what are the differences and could these affect enzymatic activity, hexamer formation etc? Are GDH1/2 on different chromosomes in corn? Are double mutants lethal? Etc. etc.
2. I think Figure 3 is one of the most important figures but must be improved and explained in more detail. In part A the molecular weights of the marker lane should be indicated. The gel and the text show GDH2 to be larger than GDH1 however in Section B GDH2 (in the WT) and the combinations with GDH1 appear to be smaller, it is not clear to me why this is the case: I suggest to label the individual hexamers in both the leaf and root gels and keep the a/b notation rather than a/b also the band denominated 7b should be 6b/6b , if I understood correctly.
Is there any relationship between the intensity of the bands in the WT and expression levels?
Part C of Figure 3 needs to be improved so that the different bands can be clearly distinguished-lane mm should be the same as lane gdh2 in part B or perhaps I misunderstood?
3. I think the effect of the heterozygote is clear, however I think the authors could have extended the work to include the GDH1 mutant line for comparison and even generated a double mutant. This may have shed light on the heterozygote effect.
Analysis of expression patterns for both genes in different tissues and across the life cycle of the plants would also have been useful even if only in the WT.
4. The Bailey et al reference is repeated in the reference list.
In general only minor corrections are necessary in terms of English language throughout the manuscript.
Author Response
- I think it would be useful to include more information in the introduction about GDH genes in general in different plant species: for example is thaliana the only species with 3 genes, what are the differences in expression patterns in maize and other species, at the amino acid level what are the differences and could these affect enzymatic activity, hexamer formation etc? Are GDH1/2 on different chromosomes in corn? Are double mutants lethal? Etc. etc.
- We thank Reviewer 3 for these constructive comments. More information has been provided as follows:
- The work on rice in which 4 genes encoding GDH have been identified is now cited and presented line 53 to 55.
- We agree that the plant amino acid content as a function of the GDH isoenzyme composition is a key issue which however remains practically unsolved due to combined species- and organ-specificities as well as environmental constraints. It is therefore rather difficult to present a consensus view on such complex biological process. Nevertheless, we have modified the text to highlight a bit more such complexity lines 6é-64. This issue was also extensively discussed lines 325 to 339 in the paragraph presenting the work on genetically modified plants for GDH gene expression and activity.
- Additional information on gdh mutant phenotype has been provided lines 70 to 72.
- We have added two sentences concerning the chromosomal location of the two genes encoding GDH in corn and the localisation of QTLs for the corresponding enzyme acivity which did not match, with relevant references.
- I think Figure 3 is one of the most important figures but must be improved and explained in more detail. In part A the molecular weights of the marker lane should be indicated. The gel and the text show GDH2 to be larger than GDH1 however in Section B GDH2 (in the WT) and the combinations with GDH1 appear to be smaller, it is not clear to me why this is the case: I suggest to label the individual hexamers in both the leaf and root gels and keep the a/b notation rather than a/b also the band denominated 7b should be 6b/6b I think to be consistent, if I understood correctly.
Is there any relationship between the intensity of the bands in the WT and expression levels?
Part C of Figure 3 needs to be improved so that the different bands can be clearly distinguished-lane mm should be the same as lane gdh2 in part B or perhaps I misunderstood?
- We have modified Figure 3 according to the reviewer comments and improved the quality of part C. There was in inversion in the different subunits composition which is now corrected. When there is only a single subunit it is either 6a or 6b We did mot quantify the corresponding gene expression as the aim of our study was mainly focused on the enzyme activity which is the most important in terms of both plant physiology and phenotypic evaluation.
- I think the effect of the heterozygote is clear, however I think the authors could have extended the work to include the GDH1 mutant line for comparison and even generated a double mutant. This may have shed light on the heterozygote effect.
Analysis of expression patterns for both genes in different tissues and across the life cycle of the plants would also have been useful even if only in the WT.
- Of-course we made several attempts to isolate a GDH1 mutant, unfortunately without success. GDH1-2 antisense plants have also been produced. However, it was impossible to produce seeds from these plants to perform further phenotypic evaluation. These points have been discussed lines 358-364.
- We did not monitor GDH1 and GDH2 gene expression in the context of our study.
- The Bailey et al reference is repeated in the reference list.
- Corrected and list of references updated with the new ones.
In general, only minor corrections are necessary in terms of English language throughout the manuscript.
- The manuscript has been checked by Prof. Peter Lea, one of the co-authors.
Round 2
Reviewer 3 Report
All the suggestions have been taken into account and/or answered in the reply.
Some very small errors remain -in places close to where new comments have been added.